# From the Ground Up: Prairies on Reclaimed Mine Land—Impacts on Soil and Vegetation

**Rebecca M. Swab [1],\*, Nicola Lorenz [2], Nathan R. Lee [2], Steven W. Culman [3]**  **and Richard P. Dick [2]**

[1]   MAD Scientist Associates, Westerville, OH 43081, USA

[2]   School of Environment and Natural Resources, The Ohio State University, Columbus, OH 43210, USA;
      lorenz.64@osu.edu (N.L.); nlee1506@gmail.com (N.R.L.); dick.78@osu.edu (R.P.D.)

[3]   School of Environment and Natural Resources, The Ohio State University, Wooster, OH 44691, USA;
      culman.2@osu.edu

\*   Correspondence: rebecca@madscientistassociates.net; Tel.: +614-818-9157

**Abstract:** After strip mining, soils typically suffer from compaction, low nutrient availability, loss of soil organic carbon, and a compromised soil microbial community. Prairie restorations can improve ecosystem services on former agricultural lands, but prairie restorations on mine lands are relatively under-studied. This study investigated the impact of prairie restoration on mine lands, focusing on the plant community and soil properties. In southeast Ohio, 305 ha within a ~2000 ha area of former mine land was converted to native prairie through herbicide and planting between 1999–2016. Soil and vegetation sampling occurred from 2016–2018. Plant community composition shifted with prairie age, with highest native cover in the oldest prairie areas. Prairie plants were more abundant in older prairies. The oldest prairies had significantly more soil fungal biomass and higher soil microbial biomass. However, many soil properties (e.g., soil nutrients, β-glucosoidase activity, and soil organic carbon), as well as plant species diversity and richness trended higher in prairies, but were not significantly different from baseline cool-season grasslands. Overall, restoration with prairie plant communities slowly shifted soil properties, but mining disturbance was still the most significant driver in controlling soil properties. Prairie restoration on reclaimed mine land was effective in establishing a native plant community, with the associated ecosystem benefits.

**Keywords:** reclamation; prairie restoration; β-glucosoidase activity; EL-FAME; soil microorganisms

## 1. Introduction

Ecosystem restoration often focuses on restoring landscapes to historic conditions [1], or to dynamic, realistic sites that provide specific, desirable characteristics and ecosystem services [2]. For heavily disturbed areas, restoration to a historic state may be impossible, and land managers often focus on alternate, more realistic goals for restoration. These goals can include increasing plant species richness, diversity, native cover [1,3,4], plant community structure and species composition [4–6], and soil health [7]. For both restoration of reclaimed mine land and tallgrass prairie restoration in the Midwest, achieving a historic state is often impossible, so alternate goals are typically the focus.

The impacts of coal mining in the Appalachian region encompass more than 600,000 ha [8]. Though previously forested, this land has largely been reclaimed since the 1970s with herbaceous species, primarily cool season exotic grasses and forbs [8,9]. Over 30 years after reclamation, these areas can still be in arrested succession, probably due to soil compaction and competitive ground covers, with non-native and invasive species (mostly grasses and shrubs) dominating the landscape [10]. Reforestation of those areas can be successful using the forestry reclamation approach (FRA) [8,11],

however this methodology is prohibitively expensive and takes decades to mature. When FRA is not feasible for mine land, prairie planting is an alternative that potentially increases diversity across trophic levels, providing better quality habitat for pollinators [12] and other wildlife, while increasing native cover on a relatively quick (3–5 year) timescale. On degraded mine lands in the area of the Midwest known as the prairie peninsula, prairie species have been utilized in mine reclamation since at least the 1970s [13–15]. These prairie plantings have been shown to effectively establish native vegetation on reclaimed mine lands [13–18], and can also provide habitat for endangered and threatened wildlife species [19].

Most of our understanding of the effectiveness of prairie restorations comes from research on previously agricultural sites. This work has shown that prairie restorations often do not achieve the same functionality as undisturbed prairies, often having lower species richness that declines over time [6,20,21], and higher levels of exotic species [3,22,23]. However, restored agricultural prairies can have similar levels of annual net primary productivity as prairie remnants [3,20,23,24]. They can also reach high scores for floristic quality [24], though rarely as high as remnants [4]. Restored prairie soils often do not have the same soil properties as remnant prairies, often with lower total nitrogen, organic matter, and total carbon [7,23,24], though [23] and [24] found the bulk density of restored prairie sites to be at or above the range of remnants, and [21] found higher N and P levels in restored versus reference prairies. Prairie restorations with prairie plant species on agricultural lands have increased total soil microbial biomass [25], soil organic matter [26], and root biomass and carbon storage with age [23]. Others [27] reported that soil microbial communities and soil carbon levels on agricultural land approached those of remnant prairies within 13 years, after prairie species were planted. Comparisons between the soil properties of prairies and cool season grasslands are less common, though there is some evidence that there may not be significant differences in soil properties between the two grassland types [18,21].

Soil health is a major concern for reclaimed mine lands and improving soil health is a focus of mine land restoration efforts [28,29]. On reclaimed mine lands, soils have often suffered from compaction, loss of soil carbon [30], an increase of geogenic organic carbon (coal carbon) [31], loss of soil structure [32,33], lower soil water potential and content [34], and disrupted soil microbial communities. Here, prairie plantings on reclaimed mine land can increase soil productivity, fertility, and soil organic matter [35,36], and might, in the long-term, ameliorate compacted soils. In comparison to shallow-rooting cool-season grasses, which are typically used in reclamation, warm-season prairie vegetation tends to have deeper roots, and thus might be beneficial to improve soil health in deeper soil layers [23]. However, another study [18] did not find differences in soil health or microbial community structure between prairies and cool season grasslands, two years after planting; but given the short time frame of that study and the typical decadal length of many soil processes, longer term studies are needed.

The potential benefits of prairie restorations on reclaimed mine lands or agricultural sites are substantial; prairies can sequester carbon [37,38], provide habitat to endemic wildlife, provide high quality nectar and pollen resources for pollinators, and serve as grazing land, especially in the summer when cool season grasslands are not productive. Therefore, there is a need for further understanding of how effective prairie remediation efforts are on Appalachian lands in achieving restoration goals, particularly focusing on increasing native cover and changing soil properties.

A wildlife conservation center, The Wilds, in southeastern Ohio has approximately 305 ha of mine land that was restored by planting warm season prairie grasses and forbs within a ~2000 ha area, where exotic cool season grasses and forbs were established in reclamation. The objective of this study was to determine the success of the prairie restorations, specifically with regards to changing the soil microbial community, increasing soil carbon, and establishing a native plant community.

## 2. Materials and Methods

*2.1. Site Description*

The study was conducted at The Wilds, a 3700 ha center for conservation, research, and education, located on reclaimed mine land in Muskingum County, OH (39.828° N, −81.732° W) in the Appalachian plain of southeastern Ohio (Supplementary Materials Figure S1). Prior to mining, this site consisted of farmlands interspersed with second-growth mixed mesophytic forest [39]. Over 90% of the land was mined for coal at various times between 1940 and 1984. In the 1940s to the early 1970s, reclamation on site consisted primarily of tree planting. Lands reclaimed between 1972 and 1977 were reclaimed as grasslands under the Ohio Reclamation law and are called "B" permit areas. B permit areas were re-seeded with a mix of predominantly *Lolium perenne* (perennial ryegrass, ~15.7 kg ha$^{-1}$), *Festuca arundinaceae* (tall fescue, 9 kg ha$^{-1}$) and *Dactylis glomerata* (orchard grass, 9 kg ha$^{-1}$). Lower rates of *Lespedeza cuneata* (sericea lespedeza), *Agrostis gigantea* (redtop), *Medicago sativa* (alfalfa), and *Trifolium pratense* (red clover) were added in the mix, as well as a few other species of grasses and legumes, and low stocking rates of *Elaeagnus umbellata* and some trees, none of which survived (Unpublished report). In 1977, the Surface Mining Control and Reclamation Act was passed. At this site, "C" permit areas were reclaimed under this act between 1977 and 1984. These areas were planted at 9–16 kg seed ha$^{-1}$, with a mix of non-native species, a majority of the mix consisting of cool season grasses *D. glomerata* and *L. perenne.* Other species such as *Lotus corniculatus* (bird's-foot trefoil), *M. sativa, T. pratense, F. arundinaceae,* and *P. pratensis* were included at lesser rates, and the exact mix varied by location. Exact date of planting was not recorded for most grasslands; mining records were incomplete, permits sometimes overlapped, and aerial photos were taken only every 10 years (Parker, Unpublished report). Thus, for this study we classified grasslands by the type of permit rather than exact date of reclamation. Those cool season grasslands described above will hereafter be called "Baseline". Areas within this study which were not mined "U" are primarily areas where mining was underground rather than at the surface, due to the height of the topography. While there are "remnant forests" within these high points, for this study only grasslands were included.

Since taking ownership of the land, The Wilds has managed the land with a variety of techniques including grazing, tree plantings, invasive species removal, and tallgrass prairie planting. Prairie plantings on site began in 1999, and by 2016 covered 272 ha. Cool season baseline areas were sprayed with herbicide, and then seeded with a mix of native warm season prairie grasses and forbs with a no-till drill. For this study a total of 15 grasslands were selected, 4 baseline areas, which were planted as cool season grasslands after reclamation, but have had no additional restoration, and 11 restored prairie grassland areas, where the baseline grasses were replaced with mixes of warm season grasses and forbs. We selected two (group A) or three (groups B, C, and D) grasslands within each group A: 14–19 years (2) B: 8–10 years C: 4 years D: 2 years X: Baseline (years refer to prairie age in 2018). Exact prairie planting varied by site but typically included primarily tall-grass warm-season prairie species, namely *Panicum virgatum* (switchgrass), *Andropogon gerardii* (big bluestem), and *Sorghastrum nutans* (Indiangrass), with legumes such as *Cassia herbecarpa* (wild senna) or *Chamaecrista fasciculata* (partridge pea), and native forbs such as *Asclepias tuberosa* (butterflyweed), *Heliopsis helianthoides* (false sunflower), *Ratibida pinnata* (gray-head coneflower). Prairies were established between 1999 and 2016, in areas which were reclaimed from mining between 1972 and 1984 (Table 1). Soil type in all grasslands was primarily Morristown silty clay loam, with some Westgate silt loam at Overlook (11.6%), Bethesda silt loam at Beetle (4.8%) and Lake trail (100%), and Lowell-Gilpin complex at Nomad (4.4%). Slopes were split about evenly between 0-8% and 8–25%, with a few locations 25–70% (Overlook, 37%, Dip, 25.6%) (Supplementary Figure S1 [40]).

**Table 1.** Characteristics, vegetation cover, and soil parameters (in 2017) for grassland study areas.

| Prairie Name | Grassland Type | Date Planted Groups [a] | Geographic Group | Reclamation Permit | Native Cover Category [b] | pH (0–15 cm) | pH (15–30 cm) | Total Organic Carbon (TOC) (15 cm) | TOC (15–30 cm) |
|---|---|---|---|---|---|---|---|---|---|
| Bison Base | Baseline | X | Central | B (1972–77) | A | 7.8 | 7.8 | 1.976 | 0.654 |
| Butterfly Habitat Base | Baseline | X | South | C (1977–84) | A | 7.7 | 8.0 | 2.315 | 1.248 |
| Butterfly Habitat Expansion | Prairie | B | South | C (1977–84) | C | 7.4 | 7.4 | 2.871 | 0.734 |
| Butterfly Habitat | Prairie | A | South | C (1977–84) | B | 6.8 | 7.8 | 1.93 | 0.888 |
| Bison Knoll | Prairie | C | Central | B (1972–77) | C | 7.8 | 7.9 | 2.39 | 0.634 |
| Beetle | Prairie | B | East | C (1977–84) | B | 7.7 | 7.7 | 2.155 | 0.725 |
| Dip Base | Baseline | X | East | C (1977–84) | A | 7.7 | 8.0 | 2.314 | 1.007 |
| Dip | Prairie | D | East | C (1977–84) | B | 7.6 | 7.6 | 2.277 | 0.998 |
| Hill 1 | Prairie | C | Central | B (1972–77) | A | 7.5 | 8.0 | 2.987 | 0.835 |
| Lake Trail | Prairie | A | Central | C (1977–84) | C | 7.1 | 7.1 | 2.568 | 2.197 |
| North Prairie | Prairie | A | North | B (1972–77) | C | 7.8 | 8.2 | 2.215 | 0.805 |
| Nomad Ridge | Prairie | C | South | Unmined | C | 7.6 | 8.0 | 1.883 | 0.393 |
| Overlook | Prairie | D | South | C (1977–84) | B | 7.1 | 7.8 | 3.178 | 0.663 |
| Zion Base | Baseline | X | North | B (1972–77) | A | 7.9 | 8.1 | 1.391 | 1.202 |
| Zion Prairie | Prairie | D | North | B (1972–77) | B | 7.8 | 7.8 | 2.243 | 0.602 |

a. Date planted (as prairie) groups: A= 14–19 years B= 8–10 years C= 4 years D= 2 years X= Baseline (shaded: cool season).

b. Native cover category: A = 6–20% B = 36–48% C = 54–93%

We recognize that our use of grasslands is subject to the limitations of pseudoreplication and space for time limitations. Prairie plantings were originally planned as a management technique, therefore grasslands were planted in different areas with different soils in different years, and not truly replicated. We addressed this issue by using specific grasslands, rather than a transect or quadrat within each grassland, as our experimental unit for analyses. Soil was aggregated within each grassland; vegetation quadrat values were averaged to generate mean values for each grassland.

In order to determine how various factors were influencing soil and vegetation characteristics, we made comparisons by grouping the sites by the following:

(a) Grassland types: Prairie, Baseline
(b) Prairie age (in 2018): A (14–19 years), B (8–10 years), C (4 years), D (2 years)
(c) Geographic location: C (Central), E (East), N (North), S (South)
(d) Permit type: B (1972–1977), C (1977–1982), U (Unmined)
(e) Native plant cover percentage: A (0–25%), B (25–50%), C (>50%)

While several of these groupings are overlapping (i.e., grassland types and native cover, geographic location, and permit type), the groupings were compared separately to improve understanding of each factor.

*2.2. Plant Sampling*

At each grassland, three 40 m transects were established using the soil sampling subareas as a center. In September 2016, 2017, and 2018, vegetation was sampled, though only in 2018 was every grassland sampled due to mowing or other disturbances in some grasslands in 2016 and 2017. Every five meters along the transects, a 1 x 1 m quadrat frame was placed, for a total of nine quadrats per transect, using a modified Daubenmire method [41]. Within each quadrat, all plant species present were identified, and their percentage cover visually estimated.

Each plant was categorized as native, naturalized, or invasive [42]. Native plants were defined as those with a pre-industrial record of presence within the Appalachian Plateau [43] and within Ohio prairies, even though the Appalachian plateau is beyond the "prairie peninsula", as mapped by [44]. Pockets of prairie may have existed within the Appalachian plateau. Naturalized plants were defined as non-native species adapted to Ohio, which do not pose a threat of being invasive [42]. Invasive plants were defined as non-native species that are able to invade native ecosystems and displace native plants [45].

Plant biomass samples were clipped, oven-dried, and weighed, following the sampling protocol outlined by [46], to estimate plant production at 11 of the grasslands in 2017 only (some grasslands excluded because of seasonal mowing). Shoots were clipped to stubble height (~7–10 cm) from within $6 \times 0.25$ m$^2$ quadrats tossed randomly along a meander across each plot. Collections were taken within a two week window beginning 10 September 2017, and dried within 48 h of clipping.

## 2.3. Soil Sampling

Within each of the grasslands, three subareas were chosen along an elevation gradient. In Spring 2017 and 2018, at each of the three subareas, a 2.5 cm soil probe was used to collect samples to two depths: 0–15 cm and 15–30 cm. Ten to 12 samples within each subarea were collected within a 10 m circle and combined by depth, resulting in 6 composite samples per site. Each of the composite samples was sieved through a 2 mm sieve and divided into two bags, one stored in a 4 °C cooler, and the other in a −20 °C freezer.

### 2.3.1. Soil Chemistry

Standard soil chemistry analyses (pH, nutrients including P, K, Fe) were conducted in the laboratory following the methods in [47]. The percentage of total organic carbon (TOC) and geogenic organic carbon (GOC) in soil samples was determined using an elemental analyzer (Carlo Erba CHN EA 1108). Before TOC analysis, 2 g of air-dried and ground soil was treated 3 times with 20 mL 1M HCl. Samples were allowed to react for 30 min. 50 mg of the HCL treated soil was separated and rinsed 5 times with H$_2$O before being analyzed for % C. Total C (TOC) and N were measured by dry combustion (950 °C) with a Vario Max CN analyzer (Elementar, Hanau, Germany).

For the analysis of geogenic organic carbon (GOC), the TOC soil was treated with nitric acid and hydrofluoric acid, and rinsed 5 times with H$_2$O, and subsequently transferred into a porcelain crucible, then placed into a muffle furnace at 340 °C for 3 h. The remaining C resistant to thermal oxidation is defined as geogenic organic carbon, and was analyzed for % C using the elemental analyzer described above [31]. Finally, soil organic carbon (SOC) was calculated by subtracting the geogenic organic carbon (GOC) from the TOC.

Particulate organic matter was determined by sieving and the loss-by-ignition method (LOI) [48]. Briefly, 30 g of 2-mm sieved soil was dispersed by adding sodium hexametaphosphate (HMP), at an aqueous concentration of 0.5% by weight, and shaking the soil sample for 16 h (overnight) on a reciprocating shaker at 120 reciprocations per minute in a container with a 3:1 HMP (90 mL) to soil (30 g) ratio. After dispersion, the soil slurry was sieved through nested standard 0.5-mm mesh (no. 35) and 0.053-mm mesh (no. 270) sieves to separate sand particles and POM. The collected particles on the 0.053 -mm sieve were dried at 55 ºC to constant weight, and then subjected to 450 ºC for 4 h to measure POM by LOI.

### 2.3.2. Soil Microbial Analyses

To determine the extracellular enzymatic activity of β-glucosidase, soil was incubated with a substrate (*p*-nitrophenyl-β-D-glucoside) and pH buffer (MUB pH6) for one hour. The product (*p*-nitrophenol 6 (PNP)) concentration was determined colormetrically [48].

Microbial community fatty acid methyl ester (FAME) profiles of soils were determined by the ester-linked method [49]. In brief, three g of soil were mixed with a 0.2 *M* KOH in methanol solution

and incubated at 37 °C for 1 h with periodic vortexing, followed by the addition of 1.0 *M* acetic acid to neutralize the solution pH. Extracted FAMEs were partitioned into an organic phase by adding 10 mL of hexane followed by centrifugation (10 min at 480 × g) to separate organic matter from the hexane layer that separated and evaporated under a stream of $N_2$. FAMEs were dissolved in 1:1 hexane: methyl-*tert*-butyl ether and analyzed on a Hewlett-Packard 5890 Series II gas chromatograph (Palo Alto, CA, USA) equipped with a fused silica capillary column (5% diphenyl-95% dimethylpolysiloxane) and a flame ionization detector. FAMEs were identified and their relative peak areas determined by the MIS Aerobe method of the MIDI system (Microbial ID, Inc., Newark, DE). FAMEs are described by standard nomenclature. Numbering of carbons begins at the aliphatic (ω) end of the molecule. The number of double bonds within the FAME is given after the colon. *Cis* and *trans* conformations are designated with suffixes "*c*" and "*t*", respectively. Other notations are "Me" for methyl, "OH" for hydroxy, "cy" for cyclopropane, and the prefixes "*i*" and "*a*" for *iso-* and *anteiso*-branched FAMEs, respectively. Fatty acid biomarkers were used to quantify various microbial groups (according to 45 and 46) as follows:

- Actinobacteria = Sum of 16:0 10-methyl, 17:0 10-methyl, and 18:0 10-methyl
- General Bacteria = Sum of 14:0 iso, 15:0, and 17:0
- Gram⁻ Bacteria = Sum of 16:1 ω7c, 17:0 cyclo, 19:0 cyclo ω8c, and 18:1 ω7c
- Gram⁺ Bacteria = Sum of 15:0 iso, 15:0 anteiso, 16:0 iso, 17:0 iso and 17:0 anteiso
- Fungi = Sum of 18:2 ω6c and 18:1 ω9c
- Arbuscular mycorrhiza fungi (AMF) = 16:1 ω5c
- Protozoa = Sum of 20:2 ω6c, 20:3 ω6c and 20:4 ω6c
- Total FAME concentration (sum of all above) served as a marker for soil microbial biomass

*2.4. Data Transformations and Statistical Analyses*

2.4.1. Microbial Data Transformations

Enzyme activity and EL-FAME are presented directly and transformed as ratio; standardized for clay content [50]. The per clay unit clay ratio was done to isolate these indicators, independently of soil type, because soil type (in particular distribution of texture) is a major controller of nearly all soil properties, which can obscure land management effects on data [51–56].

2.4.2. Data Analyses

Vegetation and soil data analyses were completed using R version 3.6.0 [57]. Graphing was completed with use of R core and the ggplot2 package [58]. Correlation of soil and vegetation factors was plotted with the package corrgram [59].

Plant species richness and Shannon–Weiner diversity were calculated using R package, vegan [60]. Quadrat values were averaged to find a mean value for each site. Non-metric multidimensional scaling (NMDS) was performed using the vegan package to demonstrate vegetation community similarity between grasslands, and the anosim function was used to perform a PermANOVA test for difference between groups. For each treatment, betadisper function, a multivariate analogue of Levene's test in the vegan package, was used to test homogeneity of variance between treatments. Metamds function mapped ordination between microbial communities. Soil microbial communities were plotted with the same a–e groupings as listed above for significance testing. Glucosidase and soil chemical parameters were not included in ordination plotting.

2.4.3. Significance Testing

Parameters were first checked for normal distribution with the Bartlett test. If normal, ANOVA was calculated, and TukeyHSD post hoc tests were utilized, all utilizing the base R program. If non-normal, Kruskal tests and post hoc Kruskal Nemenyi tests (package PMCMRplus [61]) were used to test for

significant differences. Comparisons were made for each soil and vegetation parameter within each of the a–e categories, listed above in Section 2.1.

## 3. Results

### *3.1. Plants*

#### 3.1.1. Plant Community Composition

By 2018, plant community composition showed separation with time since planting, with the youngest prairies having the most similarity to baseline areas, and increasing dissimilarity with time (Figure 1, p=0.036). Younger prairies could be characterized by higher abundances of cool season exotics such as *Lolium perenne*, *Trifolium pratense*, and *Lotus corniculatus,* whereas older prairies showed higher abundances of the planted native prairie species, such as *Helianthus maximiliani* (Maximilian sunflower) and *Andropogon gerardii*. Community composition differences were also seen with community composition by native cover and grassland type, but not geography or permit area (not shown). Individual differences in composition between sites was explained by the time since planting groupings. Aboveground plant biomass in 2017 was typically much higher in prairies (Supplementary Table S1).

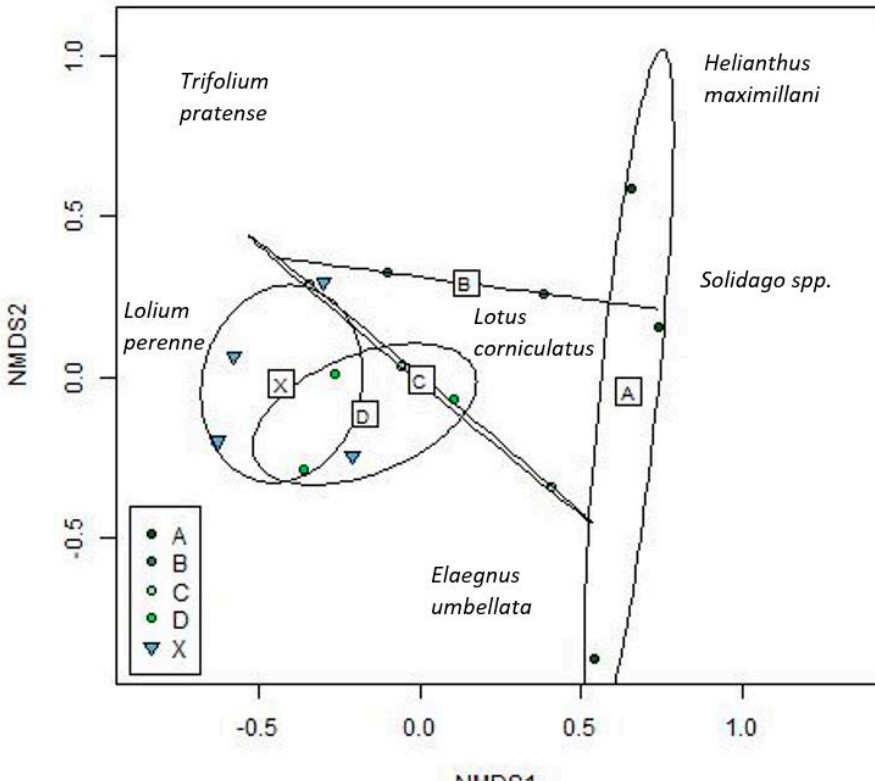

**Figure 1.** Grassland vegetation communities in 2018. Dominant plant species were chosen to represent each axis. Ellipses represent grassland groupings: (**A**): Prairie aged 14–19 years (**B**): Prairie aged 8–10 years (**C**): Prairie aged 4 years (**D**): Prairie aged 2 years (**X**): Baseline (cool season). Circles represent prairie grasslands, and triangles represent baseline cool season grasslands.

#### 3.1.2. Plant Richness and Diversity

Species richness and diversity were not found to be significantly different between individual grasslands (Supplementary Table S2), geography, permit area (not shown), or grassland type (prairie vs cool-season) in 2017 or 2018 (2018 shown in Figure 2a), but were found to be significantly different by planting date, with the youngest prairies having the highest diversity, and the oldest with the

lowest (Figure 2b). Cool season baselines had non-significantly higher plant diversity (and richness, not shown) than prairies as a group in 2018 (Figure 2a). Native cover increased with prairie age and was significantly higher in prairie versus baseline areas (Supplementary Table S2, Figure 3). Cover of individual plant species by grassland for 2018 is in Supplementary Table S3.

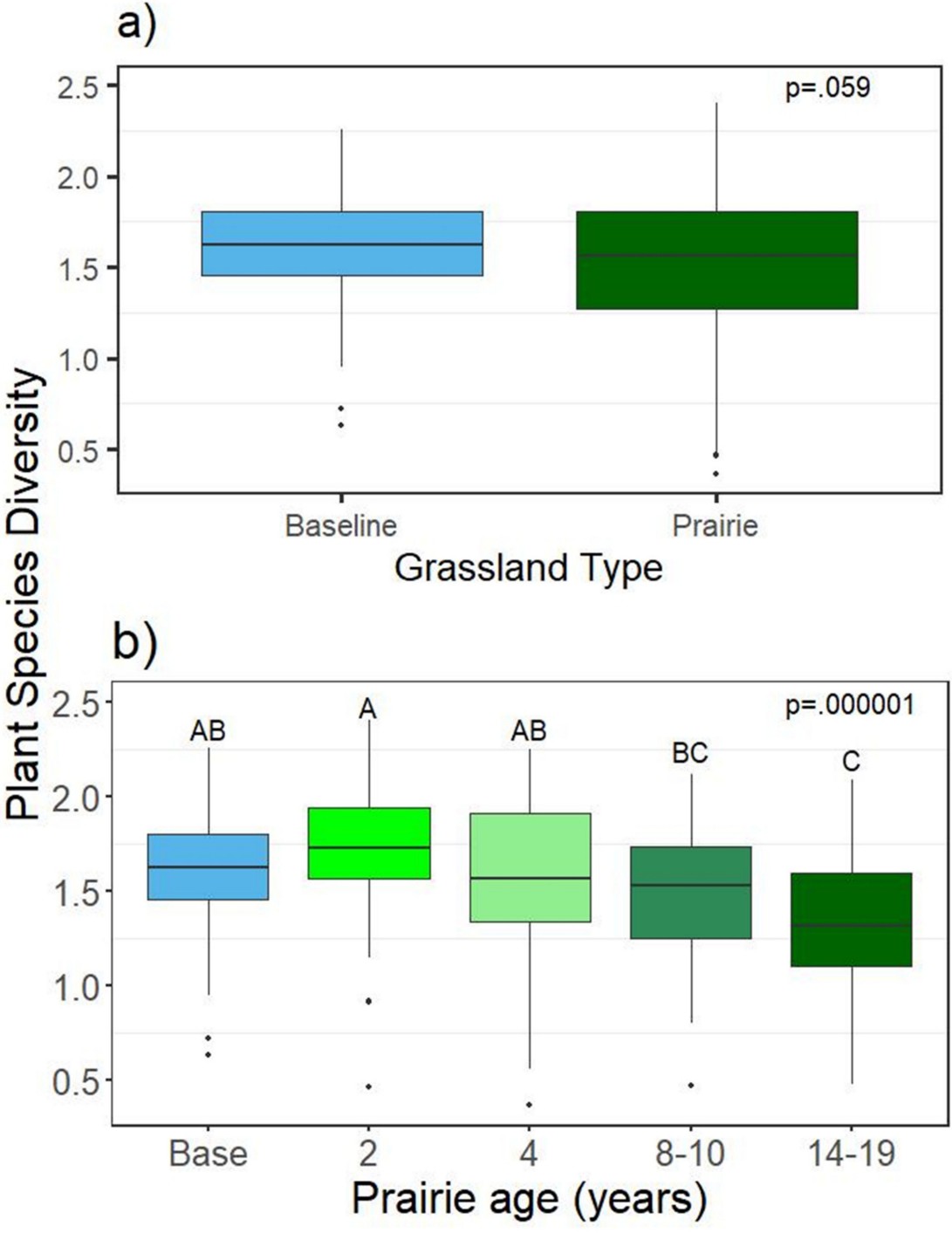

**Figure 2.** Average Shannon–Wiener plant species diversity in grasslands in 2018. (**a**) Grassland type, (**b**) prairie age. Baseline sites are cool season grasslands.

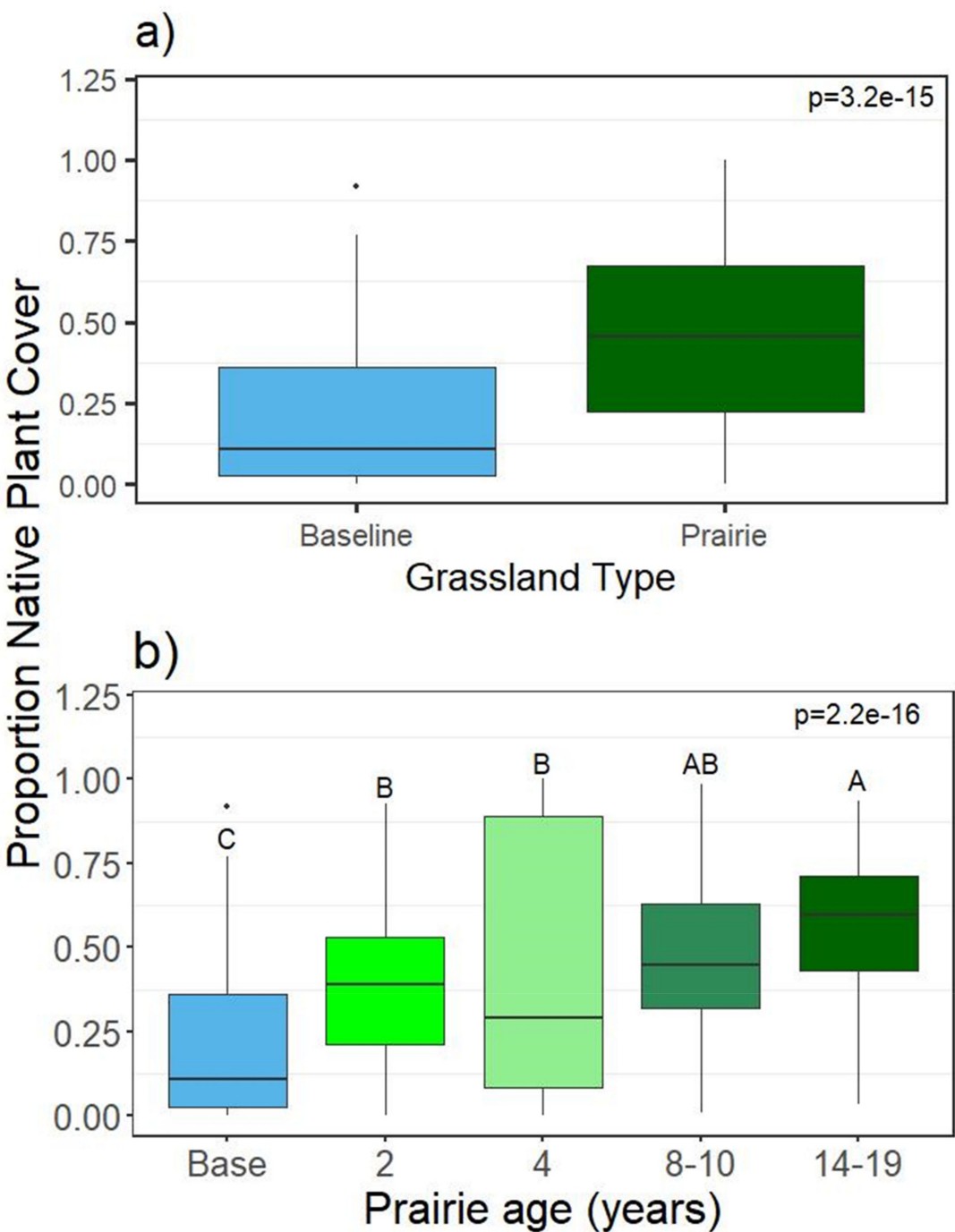

**Figure 3.** Average proportion of native plant cover in grasslands in 2018, shown by (**a**) grassland type and (**b**) prairie age. Different letters indicate statistical differences between treatments ($p < 0.05$). NP = baseline (not planted to prairie: cool season).

*3.2. Soil*

3.2.1. Soil Chemistry

Soil pH values were mostly alkaline, ranging from 6.8 to 8.1 (Table 1), with a median of 7.7. Soil organic carbon (SOC) ranged from 0.4% to 2.4% in 0–15 cm soil depth, and 0% to 1.3 % in 15–30 cm soil depth (Table 2). Soil geogenic organic carbon (GOC), ranged from 0.48% to 1.1% in 0–15 cm, and from 0.35% to 0.9% in the 15–30 cm, soil depth (Supplementary Table S1). On average, total organic carbon (TOC) consisted of 46% geogenic organic carbon (Supplementary Table S1). Soil organic carbon

(SOC, Table 2), geogenic organic carbon (GOC), total organic carbon (TOC), and particulate organic matter (POM) were not significantly different between grassland type or prairie age (GOC, TOC, POM, groupings not shown).

**Table 2.** Soil fungal abundance, microbial biomass in 2018, and soil organic carbon (SOC) in 2017 classified by factors in the experimental design at 0–15 cm and 15–30 cm depth.

| | 0-15 cm Soil Depth | | | | | | 15-30 cm Soil Depth | | | | | |
|---|---|---|---|---|---|---|---|---|---|---|---|---|
| | Total Fungi (n mol g$^{-1}$ clay) | | Total El-FAME (n mol g$^{-1}$ clay) | | SOC (%) | | Total Fungi (n mol g$^{-1}$ clay) | | Total El-FAME (n mol g$^{-1}$ clay) | | SOC (%) | |
| **Grassland type** | | | | | | | | | | | | |
| *p value* | **0.016** | | *0.015* | | *0.133* | | *0.040* | | *0.029* | | *0.59* | |
| **Baseline (cool season)** | 180.24 | A[1] | 483.46 | A | 1.22 | | 38.80 | A | 87.13 | A | 0.33 | |
| **Prairie** | 232.66 | B | 607.82 | B | 1.62 | | 54.44 | B | 118.30 | B | 0.22 | |
| **Prairie age (in 2018)** | | | | | | | | | | | | |
| *p value* | *0.006* | | *0.065* | | *0.454* | | *0.0008* | | *0.0004* | | *0.515* | |
| **A (14–19 years)** | 282.70 | A | 667.53 | | 1.37 | | 77.94 | A | 155.75 | A | 0.50 | |
| **B (8–10 years)** | 225.81 | AB | 566.45 | | 1.77 | | 55.22 | AB | 116.51 | AB | 0.19 | |
| **C (4 years)** | 214.87 | AB | 626.01 | | 1.58 | | 47.57 | AB | 112.29 | AB | 0.02 | |
| **D (2 years)** | 204.97 | AB | 557.49 | | 1.81 | | 37.29 | B | 88.04 | B | 0.15 | |
| **X (Baseline- not planted)** | 180.24 | B | 483.46 | | 1.22 | | 38.80 | B | 87.13 | B | 0.33 | |
| **Geographic location (on site)** | | | | | | | | | | | | |
| *p value* | *0.109* | | *0.009* | | *0.118* | | *0.20* | | *0.025* | | *0.746* | |
| **Central** | 235.55 | | 679.31 | A | 1.57 | | 56.70 | | 130.69 | A | 0.37 | |
| **East** | 228.19 | | 583.67 | AB | 1.82 | | 41.66 | | 94.82 | AB | 0.30 | |
| **North** | 171.05 | | 468.49 | B | 1.40 | | 40.98 | | 80.97 | B | 0.15 | |
| **South** | 227.95 | | 526.25 | AB | 1.05 | | 55.86 | | 119.94 | AB | 0.10 | |
| **Mining Permit area** | | | | | | | | | | | | |
| *p value* | *0.83* | | *0.63* | | *0.28* | | *0.34* | | *0.18* | | *0.19* | |
| **B (1972–1977)** | 221.65 | | 588.93 | | 1.30 | | 42.50 | | 93.24 | | 0.08 | |
| **C (1977–1982)** | 217.02 | | 565.21 | | 1.69 | | 56.23 | | 123.14 | | 0.40 | |
| **Unmined** | 214.16 | | 564.51 | | 1.39 | | 49.26 | | 105.28 | | 0.04 | |
| **Proportion Native Vegetation in Grassland** | | | | | | | | | | | | |
| *p value* | *0.003* | | *0.03* | | *0.6* | | *0.002* | | *0.003* | | *0.788* | |
| **A 0–25%** | 184.32 | A | 513.64 | | 1.35 | | 40.13 | A | 92.35 | A | 0.26 | |
| **B 25–50%** | 209.73 | A | 555.53 | | 1.64 | | 44.49 | A | 97.68 | A | 0.16 | |
| **C >50%** | 261.98 | B | 654.80 | | 1.55 | | 66.18 | B | 139.93 | B | 0.32 | |

Means followed by A,B letters denote significant differences among groups (shaded cells). El-Fame: Ester-Linked Fatty Acid Methyl Ester, enzyme produced by soil microbes.

### 3.2.2. Soil Microbes

Most soil properties, including soil microbial ester-linked fatty acid methyl ester (El-FAME) biomarker concentrations (Table 2), and β-glucosoidase activity (Supplementary Table S4) were present in higher levels in the upper, 0–15 cm soil depth than the lower, 15–30 cm soil depth in 2017 and 2018. Prairie grasslands showed higher levels of most soil microbial groups compared to baseline grasslands in the 0–15 cm soil depth, and some differences for the 15–30 cm soil depth (Supplementary Table S4), when scaled by clay content. Within each soil depth, there was significant overlap in microbial community composition (EL-FAME data) between years (Figure 4a,b), therefore we focused on the 2018 data for the remaining figures. Given the history of mining on the site, which began in the north and worked its way south, and with changes to methodology occurring with changes

in permit laws (From B to C), and due to gradual changes in operators and techniques with time, we explored whether microbial communities were influenced by these changes. Some clustering of the soil microbial community composition was seen with permit type and geographic area within The Wilds, which varied by time of reclamation (Figure 4c–f). Arbuscular mycorrhizal fungi (AMF) were more influential on community composition in soils of the southern area (Butterfly Habitat, Butterfly Habitat Expansion, Nomad, and Overlook), in both soil depths (Figure 4f), though there was not a significant difference in their abundance by geography (Supplementary Table S4).

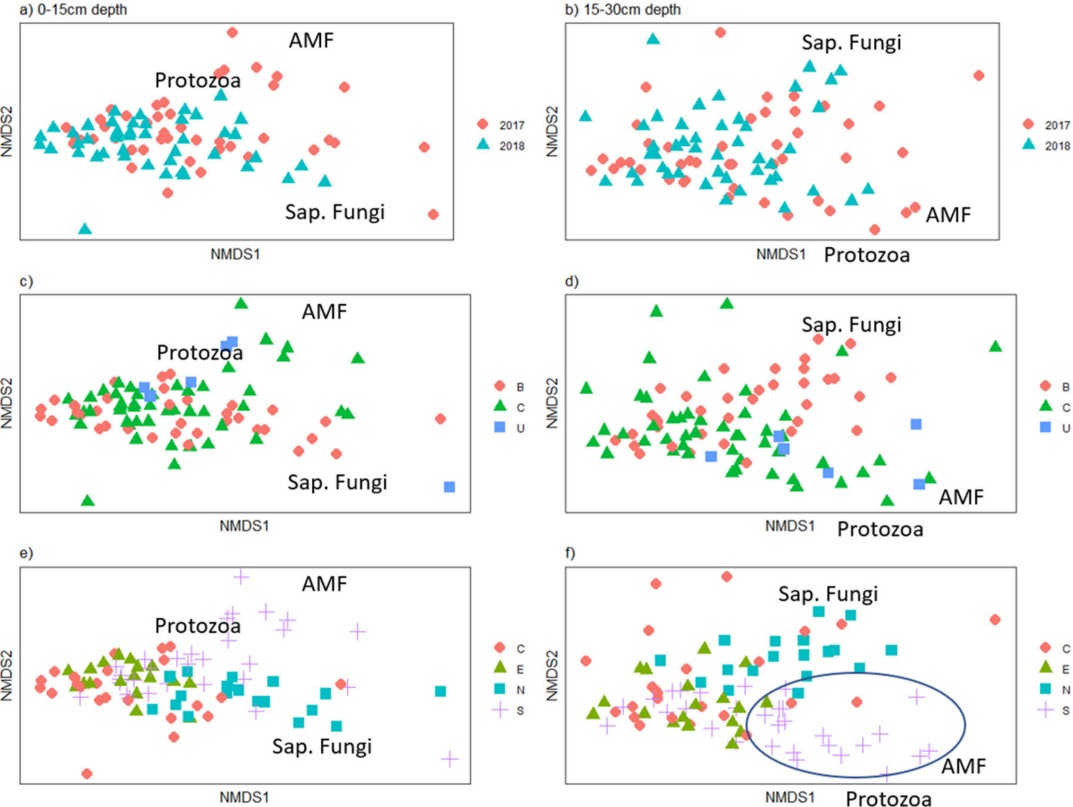

**Figure 4.** Soil microbial community composition in grasslands in 2017 and 2018. Shown for the 0–15 cm soil layer (left- **a**, **c** and **e**), and the 15–30 cm layer (right- **b**, **d** and **f**). Plotted by: Year: 2017 vs 2018 (Top row, **a** and **b**). Permit area: **B**: 1972–1977 **C**: 1977–1982 U: Unmined (Second row, **c** and **d**). Geographic location within site: C: Central E: East N: North S: south (bottom row, **e** and **f**). Sap. Fungi- Saprophytic fungi. AMF- arbuscular mycorrhizal fungi.

Some differences in both individual and overall microbial values were found between grassland type, prairie age, geography, and percentage of native cover for both soil depths when scaled by clay content, with more differences in the 0–15 cm layer for grassland type and geography, and more differences in the 15–30 cm soil layer for geography and native proportion (Supplementary Table S4). Differences between grassland type and prairie ages were largely driven by fungi, with a corresponding shift in the fungal bacterial ratio. Soil microbes tended to be positively correlated with each other but had weak and mixed relationship with soil properties such as pH, nutrients, and organic carbon (Figure 5). Soil microbes and properties had weak/mixed relationships with aboveground cover (Figure 5).

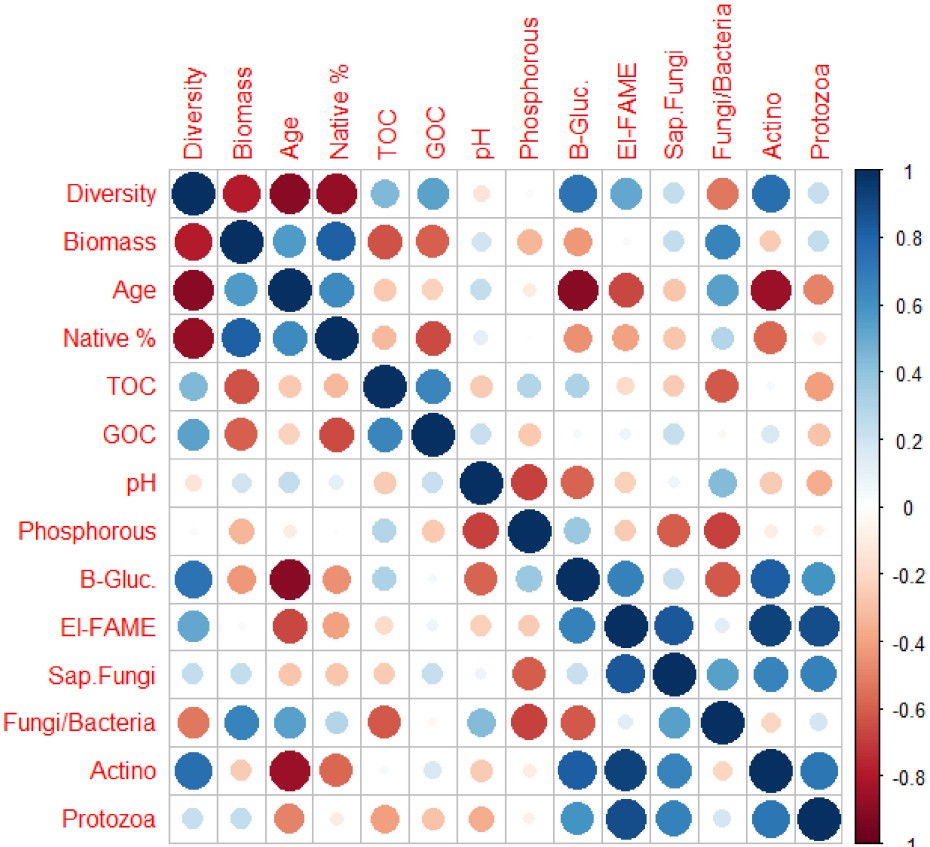

**Figure 5.** Correlations between vegetation, soil nutrients, and soil microbial data in 2018. Red circles = negative relationship. Blue circles = positive relationship. Larger circles = stronger relationship. Age: time since prairie restoration; Native %: vegetation native percentage cover; TOC: total organic carbon; GOC: geogenic (coal) carbon; B-Glu: betaglucosoidase; Sap. Fungi: saprophytic fungi; Fungi/Bacteria: fungal/bacterial ratio; Actino: actinobacteria.

Initial analyses showed no significant differences in enzyme activity or EL-FAME biomarkers for either soil depth between grassland type, regardless of age. However, when standardized to clay content (microbial biomarkers divided by clay content at each site and then multiplied by 100), most of these biochemical and microbial indicators were significantly higher in the prairies versus baseline site at the 0–15 cm soil depth; especially saprophytic fungi, total fungi, and gram were significantly higher in prairies over the baseline site in the 0–15 cm surface soil depth (Supplementary Table S4). Prairies in the northern area (North Prairie (NP), Zion Prairie (ZP)) in particular had higher microbial biomarkers for fungi (Table 2). When standardized for clay content, there were also a number of significant differences by: proportion of native plants, with grassland sites with highest proportion of natives having the highest fungal biomarker concentrations; and geography, with the central grasslands having the highest levels of many microbial properties. For both soil depths, grasslands with more than 50% native cover had higher amounts of microbes than grasslands with 0–25% or 25–50% native cover. For the 15–30 cm depth many of these differences were significant (Supplementary Table S4).

## 4. Discussion

This study showed that reclaimed mine land can successfully be converted from cool season grasses to prairie grasses in the Appalachian region, though not all restoration goals may be met in less than 20 years. Replacing cool season grasses with prairie species on reclaimed mine land: (a) increased native plant cover; (b) increased fungal and other microbial biomarkers in soil, as well as total microbial biomass; (c) trended (non-significantly) to increase SOC in the deeper soil layers

(15–30 cm); but (d) did not increase plant species richness and diversity; and (e) did not significantly increase soil organic carbon. Given these changes, it seems likely that reclaimed mine soil will increase in soil health properties more quickly when planted with prairies over cool season grasses, however it will still be a slow and gradual process.

*4.1. Plants: Richness, Diversity, and Cover*

Though plant species richness and diversity were higher in younger prairies than cool season baselines, this declined with time since restoration, which is commonly found in restored prairies [6,20,21,24]. As C4 grasses become dominant, this is typically at the expense of subdominant native forbs [6,20,21,24]. There was a tendency for increased grass dominance with time, where the oldest prairies had higher plant biomass and native cover, lower plant diversity and richness, and a different community composition. This included some specialist prairie species such as *Helianthus maximillani*, but dominant were the C4 deep-rooting grasses, primarily *Panicum virgatum* and *Andropogon gerardii*. Dominance has been shown to be a factor limiting diversity in many ecosystems, especially prairies, where one or two species often account for 40% to 80% of vegetation cover [62], and likely reduce the plant diversity of prairies at these study sites. Management, such as over-seeding, mowing [63], frequent fires [24], and increased seed density at time of planting [64], can all increase plant diversity in prairie restorations.

Native plant cover increased with prairie age on a timeline similar to other prairie restorations; for example, one study [23] found prairie plants increased cover from 10% to 80% within four years after planting. Without planting, natives were not increasing in the cool season grasslands; the amount of native cover in the baseline plots was similar to what another study [10] found on the same site. Ten years later, the reclaimed grasslands are still in a state of arrested succession, so planting natives in these areas is necessary in order to increase native cover. The only exception to this was the baseline site grazed by bison. Here, native plants increased in cover, apparently due to the grazing and hoof disturbance that may have allowed opportunities for the native plants to spread. A study [65] found warm season grass establishment within cool season grass areas increased significantly where cool season grass removal occurred, over areas where seeds were added but cool season grasses left intact. Grazing has been shown to increase abundance of rare species as well as plant diversity [62]. So, if grazing occurs on reclaimed grasslands, and there are local native seed sources available, this may facilitate conversion of these cool-season areas to prairies, as happened at one grassland at this site. This conversion is beneficial in multiple ways, for instance, several of the prairie plants established are known for their deep rooting properties, notably tall grass species such as *Panicum virgatum* and *Andropogon gerardii*, while others are native forbs known for their high-quality pollen and nectar resources, some of which support native pollinators such as the sun flower miner bee *Andrena helianthi*, which specializes on the *Helianthus* genus. Most of the prairie species planted provide better ecosystem services than the non-native species which they can replace.

*4.2. Soil Biochemical and Microbial Properties after Prairie Establishment*

When standardized for clay content, soil microbial EL-FAME biomarkers (in particular fungal) were found to increase with prairie establishment with increased time since restoration; similarly to [23], who found microbial biomass N and C to increase with time since prairie restoration in disturbed agricultural sites. Standardization of enzyme and microbial data based on clay content increased the separation between treatments, which resulted in greater sensitivity to detect statistically significant management effects. These results provide support to use this standardization step to enable enzyme activity and EL-FAME biomarkers to measure soil health independently of soil type. Supporting this is unpublished β–glucosidase activity data from a 20 year old native prairie site (previously under agricultural management) in Marion County, Ohio, which had enzyme activities nearly the same as similarly aged prairies in this study, despite each being on different soil types (Nicola Lorenz, personal communication, 2020). The extracellular enzyme, β–glucosidase, is the final step in

degradation of cellulose that releases glucose, an important energy source for the microbial community. Thus, it is a critical component of the C cycle and the ecology of microbial functions in soil. Enhanced β–glucosidase activities under prairies would indicate there is, cumulatively, greater potential to drive decomposition of plant cellulose, and is an indicator of a functional shift of the microbial community; which in our case corresponded to the distribution of plant species in driving this shift.

Soil microbial communities have been shown to change between ecosystem types (Harris 2009, Mendes et al. 2015). In our case, comparison of cool season grassland to prairie found microbial biomarkers to be higher in prairies, especially within the prairies with the highest native cover. Increased abundance of arbuscular mycorrhiza fungi (AMF) in particular points towards healthier soils supporting AMF establishment, which in return supports healthier plants under prairie, because AMF enable plants to acquire nutrients more efficiently, especially phosphorus. Thus, AMF will improve plant growth over the long term, which potentially leads to more plant biomass production, SOC accumulation, soil aggregation, and less erosion, due to increased water infiltration and drainage. The caveat to our results is that the AMF marker used in this study can also pick up bacterial FAMES. However, there were generally weak (and sometimes negative) correlations between AMF and Gram+, Gram-, and total bacteria, so it is likely that our results are a good indicator of how AMF was responding.

### 4.3. Soil Chemical Properties after Prairie Establishment

Soil chemical properties did not change with prairie establishment when compared with baseline cool season grasslands. Soil development such as organic matter accumulation and aggregate formation are slow processes, often occurring over decades or centuries [66]. Geogenic, or coal carbon, is still a significant component of the soil, with 27% to 71% of total organic carbon in grassland sites being coal C (Supplementary Table S1). This would indicate that degradation of coal C is slow, which is consistent with reclaimed coal mining sites in Germany where up to 91% of the total SOC in the 0–5 cm depth was geogenic carbon 20–30 years after reclamation [67,68].

Most prairie restoration studies focus on conversion from agriculture [7,20,21,69]. Despite obvious differences in soil conditions (tilled in agriculture versus completely dug up and replaced in mining), some similar results have been found in restorations on both reclaimed mine land and previously farmed lands; on agricultural land, no differences in SOM and N were found with prairie restoration when compared with cool season grasses [21], while another study [18] found no differences in soil properties between cool season and prairie restorations two years after planting on reclaimed mine land. Soil organic matter and nitrogen typically increase after cessation of farming and during primary succession [69], and few studies have compared cool season grasses to prairies, so it can be difficult to determine which soil improvements are due to successional processes, and which are due to prairies specifically. However, while differences in soil carbon were not significant, there is evidence of increases in soil carbon for the oldest prairies in this study (1.99% in cool season baselines vs 2.43% in prairies, $p = 0.11$, Table 2). With more sampling or more time, this is likely to be significantly different, especially given than 8 out of 11 prairies sampled were less than 10 years old. In other studies, root biomass of restored prairies was found to be similar to remnants after 12 years [23]. A native prairie in Marion County, Ohio had much higher SOC (2.8%) compared to SOC concentrations measured in this study (Nicola Lorenz, personal communication, 2020), indicating that prairies in this study will likely continue to increase SOC levels with time, though it may take longer than prairies on agricultural lands. Although prairie grass roots can extend up to 1 m deep into the soil profile, the majority of root biomass (70–80%) resides in the surface 20–30 cm [23], so benefits of the increased root biomass due to prairie restorations may be expected to be higher in the upper (0–15cm) soil layer.

### 4.4. Prairie Soil Development over Time

As these prairies develop, further changes in soil properties may occur. One study [7] estimates that it takes about 19–24 years after conversion from agriculture to prairie before vegetation, hydrologic, and soil biogeochemical processes stabilize; this could be longer on reclaimed mine land. However, [70]

showed root growth and mycorrhizal associations increase the first five years after prairie restoration, and remain constant thereafter. In our case the oldest prairies were around 18 years old, but most were less than 10 years, so some processes may have stabilized, but others may be in flux. However, when standardizing for soil based on clay content, positive enzyme activity, and microbial community, shifts were detected at grasslands reclaimed with prairie plant species. Extracellular enzyme activities, e.g., β–glucoisoidase was stabilized in the soil matrix, especially on soil organic matter and clay particles [71]. So when standardizing the effect of clay particles, microbial activities in prairie soils, which indicate stabilization of enzyme in the soil organic carbon fraction, are expected to increase over time. Thus, it is likely that even more enzymes will be produced by soil microorganisms in the future, and will be stabilized, which in return has positive effects on soil health. Overall the results would suggest longer periods are needed after vegetative reclamations are implemented to show major shifts in soil properties. For instance, one study found soil organic C to decrease the first two years after a grassland restoration, increase for a few years, and decrease again between 15–24 years after restoration [7].

Legacy impacts from mining may still have a strong influence on microbial communities; re-colonization of sterilized soil was shown to be strongly influenced by the order of colonization of microbial species or groups [28]. Only the oldest prairies (14–19 years old) had significant differences in fungal amounts, likely an explanation for why a similar study using prairie plants directly in reclamation found no differences in soil properties or microbial communities 2 years after reclamation [18]. In our study, grasslands were reclaimed 30–40 years ago, and had been restored to prairie 14–19 years before differences in microbial community were observed. The organic C pool increases with time since mining, even at sites not planted with prairies [71–73].

Geography of the site had an impact on soil properties, for instance the southern grasslands were more characterized by AMF fungi than other sites (ellipse on Figure 4f). At this site, geography is equivalent to changes in time since mining, with more northern areas reclaimed first (1970s) and southern areas later (1980s). Over this time period, changes in mining regulations occurred in 1977, and other, more gradual changes occurred as operators improved techniques. Changes in land use and duration of mining have been found to affect soil chemical properties [73]. A comparison of three mined sites in Ohio found macroaggregate stability not related to time since reclamation, but instead there was a large amount of variability between and within sites [74]. Therefore, soil properties of sites are likely to still be more influenced by reclamation methods or bedrock material, rather than vegetation.

## 5. Conclusions

Replacing cool season grasses with prairie species on reclaimed mine land was successful in establishing a mostly native plant community, increased soil fungal, and other microbial biomarkers, and trended to slightly increase SOC in the deeper soil layers (15–30 cm). Although most trends seen were not significant, future research will be necessary to evaluate soil health improvement by prairie vegetation. Therefore, we recommend that prairie species be planted on reclaimed mine lands, either as a conversion from cool season grasslands, or during reclamation itself, which can also be successful. Reclaimed prairie may not be as diverse or have the same ecosystem services as undisturbed prairies, however they increase native cover and ecologically valuable plant species, which have been shown to provide better pollinator resources, increase soil fungal biomass, and improve the habitat for other wildlife. Prairie is thought to require decades to reach a stable state when planted on agricultural land, therefore on reclaimed mine land the timeframe may be even longer; but in the meantime, benefits are still present. Future research will need to be conducted to elucidate if roots of native prairie vegetation are able explore the soil profile more intensely and improve soil quality of reclaimed mine soils over cool season grasslands, especially at deeper soil depths.

**Supplementary Materials:** The following are available online at http://www.mdpi.com/2073-445X/9/11/455/s1. Supplementary Figure S1: Map of all grassland sampling locations. Dates established indicate date of prairie restoration. Inset: map of location of The Wilds within Ohio, Supplementary Table S1: Individual grassland site history and soil characteristics, Supplementary Table S2: Vegetation results by grassland in 2018, Supplementary Table S3: List of plant species found and their average percent cover in each grassland in 2018, Supplementary Table S4: Table of averages and significant differences for soil microbial factors by various groupings for (**a**) 0–15 cm: upper soil layer, and (**b**) 15–30 cm: lower soil layer.

**Author Contributions:** Conceptualization, R.M.S. and N.L.; data curation, R.M.S.; formal analysis, R.M.S. and S.W.C.; investigation, N.R.L. and S.W.C.; methodology, R.M.S. and N.L.; project administration, R.M.S.; resources, R.P.D.; supervision, R.M.S., N.L. and R.P.D.; visualization, R.M.S.; writing—original draft, R.M.S.; Writing—review & editing, R.M.S., N.L., S.W.C. and R.P.D. All authors have read and agreed to the published version of the manuscript.

**Funding:** This research received no external funding.

**Acknowledgments:** We would like to acknowledge Alexys Nolan, Patrick Boleman, and numerous Wilds interns and Americorps for field work assistance. Additionally, we would like to acknowledge the NRCS EQIP and WHIP programs for funding to establish the prairies, and Nicole Cavender and Shana Byrd for their work obtaining these grants and installing the prairies. Abbey Henson and Kyle Sklenka are acknowledged for their laboratory work analyzing POM and TOC/GOC measures, respectively. Numerous students from OSU soil class ENR 5266 in 2016 and 2017 are also acknowledged for their assistance with soil preparation and analysis.

**Conflicts of Interest:** The authors declare no conflict of interest.

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
