# Peer review of "From the Ground Up: Prairies on Reclaimed Mine Land—Impacts on Soil and Vegetation"

_land, doi:10.3390/land9110455_

Round 1
Reviewer 1 Report
Please explain more thoroughly the results shown in Figures 1 and 4. Also list the common names of the plant species, to make the paper more accessible for those with less botanical background. Describe the significance of the beta-glucosoidase activity.
There are some grammatical edits to be made, most notably:
Line 437, delete "by" from the phrase "...for soil by based on clay content"
Line 451, delete "to a" from the phrase "...a similar to a study using..."
Reviewer 2 Report
See attached.

Reviewer 3 Report
Dear authors, I have read with interest your work. The idea is very good for the current applications in land reclamation and restoration. There are some changes that need to be addressed in order to improve your article.
Introduction
Change the last paragraph and clearly state your hypothesis and each of the targeted parameter related to restoration. It will help the entire article.
L177 Fatty acid methyl ester (FAME)
L230-231 Rewrite the entire sentence. Or delete it. You can explain after the first results that 2018 is more stable than 2017 for the extraction of good conclusion. In Mat and meth section just say that results present both 2017 and 2018 parameters. Otherwise, explain which results are in 2017&2018, and which one present just one year.
L234 [63] you have already cited the package.
L240 Significance testing - which package from R have you used?
L244 a-e or a-e just to make it more discoverable in the text.
Results
3.1. Plants - expand your results. In this form you have only 10 lines of text and 5 pages of graphs and tables. It will be better to present the results before each graph/table and it will make the text easier to understand.
A suggestion - Reduce the size of tables and make them specific for each subsection. Otherwise, the reader will have to go back to see your tables.
L313 Add a color legend with the interval -1 to +1, or write the value of correlation on correlogram. Any of these action will make the reader to understand the value of correlation.
Disscussion section
Do you think that is necessary to send the reader back in the result section by using (Figure 5), (Figure 1- A and B groups), (Figure 1)? Just focus on discussions.
4.1. Vegetation Richness, Diversity, and Cover - this subsection is related to 3.1. Plants. So change the name of one of them to be identical. You have almost one page for this subsection so makes necessary to expand the result subsection. Use the same suggestion for each subsection in the result vs discussion sections.
Conclusion
Do not cite in the conclusion. Rewrite them according to your results.
Round 2
Reviewer 3 Report
Dear authors, I have seen the new form of your article and have a lot of improvements. Now your work is more readable and provide a clear view on the topic.